# An *ex vivo* rat trachea model reveals abnormal airway physiology and a gland secretion defect in cystic fibrosis

**Elex Harris**[1,2], **Molly Easter**[2], **Janna Ren**[1], **Stefanie Krick**[1,2], **Jarrod Barnes**[1,2], **Steven M. Rowe**[1,2,3]*

1 Gregory Fleming James Cystic Fibrosis Research Center, Univ. of Alabama at Birmingham, Birmingham, AL, United States of America, 2 Department of Medicine, University of Alabama at Birmingham, Birmingham, AL, United States of America, 3 Departments of Pediatrics and Cell Developmental and Integrative Biology, University of Alabama at Birmingham, Birmingham, AL, United States of America

* smrowe@uab.edu

**Data Availability Statement:** All relevant data are within the manuscript and its Supporting Information files.

## Abstract

Cystic fibrosis (CF) is a genetic disease hallmarked by aberrant ion transport that results in delayed mucus clearance, chronic infection, and progressive lung function decline. Several animal models have been developed to study the airway anatomy and mucus physiology in CF, but they are costly and difficult to maintain, making them less accessible for many applications. A more available CFTR$^{-/-}$ rat model has been developed and characterized to develop CF airway abnormalities, but consistent dosing of pharmacologic agents and longitudinal evaluation remain a challenge. In this study, we report the development and characterization of a novel *ex vivo* trachea model that utilizes both wild type (WT) and CFTR$^{-/-}$ rat tracheae cultured on a porcine gelatin matrix. Here we show that the *ex vivo* tracheae remain viable for weeks, maintain a CF disease phenotype that can be readily quantified, and respond to stimulation of mucus and fluid secretion by cholinergic stimulation. Furthermore, we show that *ex vivo* tracheae may be used for well-controlled pharmacological treatments, which are difficult to perform on freshly excised trachea or *in vivo* models with this degree of scrutiny. With improved interrogation possible with a durable trachea, we also established firm evidence of a gland secretion defect in CFTR$^{-/-}$ rat tracheae compared to WT controls. Finally, we demonstrate that the *ex vivo* tracheae can be used to generate high mucus protein yields for subsequent studies, which are currently limited by *in vivo* mucus collection techniques. Overall, this study suggests that the *ex vivo* trachea model is an effective, easy to set up culture model to study airway and mucus physiology.

## Introduction

Cystic fibrosis (CF) is a genetic disease caused by mutations in the cystic fibrosis transmembrane conductance regulator (CFTR) gene, which encodes a channel responsible for the transport of chloride and bicarbonate ions across the epithelial apical cell surface [1, 2]. CFTR mutations lead to impaired ion transport, resulting in mucus stasis in multiple organs. In the

**Funding:** This work was supported by the NIH (www.nih.gov) F31HL164005-02 to ESH, 1R35HL135816-01 and P30DK072482 to SMR, R01HL152246 to JWB, and CF Foundation Grant (www.cff.org) ROWE19RO to SMR. The funders had no role in study design, data collection and analysis, decision to publish, or preparation of the manuscript.

**Competing interests:** The authors have declared that no competing interests exist.

respiratory tract, a defective CFTR results in viscous, adherent mucus, which leads to chronic infection, progressive organ decline, and early mortality [2, 3]. Although there have been many advances in the field of CF, limitations in the accessibility of animal models used to study the airway microanatomy and mucus physiology remain a challenge [4, 5].

Studying the anatomy and mucociliary physiology of the airway in intact airways is crucial for understanding the pathophysiology of CF lung disease and developing therapeutics, including genetic therapies in the pipeline. Historically, CFTR mutant mouse models have been the most widely used model to study CF pathogenesis *in vivo* [6–8]. CF mouse models have proven beneficial for advancing the field of CFTR gene therapy, acute inflammation, and transient infection, but they are limited in utility to study mucus biology and mucociliary clearance due to their failure to develop CF-like lung disease and their lack of submucosal gland development, an important source of CF mucus within the airways [7, 9, 10]. There have been multiple models developed that more effectively recapitulate the CF mucus pathophysiology observed in humans such as the CF piglet [11, 12] and CF ferret [13] models. Although these species have proven useful for studying mucus pathophysiology and to advance therapeutic strategies [9, 12, 14], they can be costly and difficult to maintain [4]. Acquisition of CF mucus not contaminated by secondary infection also remains a research priority and is difficult to address in small and medium sized animals and acquisition of sterile mucus is not viable in humans with chronic infection.

The CFTR$^{-/-}$ rat model has been developed as a more accessible rodent model of CF lung pathogenesis [15]. Unlike murine models of CF, the CFTR$^{-/-}$ rat develops airway mucus defects including submucosal gland plugging, airway fluid depletion, and mucociliary defects observed in human airways [15, 16]. Mechanistic studies of airway mucus physiology and pharmacologic interventions utilizing the CFTR$^{-/-}$ rat model have proven successful [16–19]. However, consistent and/or targeted pharmacologic delivery to the airway and longitudinal assessments of *in vivo* airway physiology remain a challenge for this model and others. Current pharmacologic delivery techniques to *in vivo* airways such as intravenous or intraperitoneal injection, oral gavage or intratracheal instillation may lead to off target effects and insufficiency/inconsistency of targeted drug delivery. Furthermore, assessing longitudinal changes of *in vivo* airways remains a challenge, due to the invasiveness of most data collection methods. Thus, proximal endpoints that operate at the tissue level could be informative, particularly for those that affect complex physiology of the mucosal surface that can be more readily interrogated after excision.

An *ex vivo* model of the airway would provide a useful tool to study mucus and mucociliary physiology in a tissue culture environment, where dosing times and concentrations may be well controlled, and tissue can be longitudinally assessed over an extended period. In this study, we present the development and characterization of a novel *ex vivo* model of CF using CFTR$^{-/-}$ rat trachea explants, cultured and maintained on a porcine gelatin matrix [20]. The CF rat *ex vivo* trachea model remains viable for weeks and recapitulates the mucus defects characterized in the *in vivo* CFTR$^{-/-}$ rat. Furthermore, the *ex vivo* trachea model responds to extended pharmacological treatments that are not feasible on freshly excised trachea. In addition, this model system can be used to collect large quantities of mucus over time or in intervals, which is a limitation of fresh tracheae. Overall, this study suggests that the *ex vivo* trachea model may be useful to study airway, mucus, and mucociliary physiology in a controlled culture environment that may be ancillary to or help circumvent some of the known limitations of current *in vivo* models.

# Materials

## Media conditions

i. PneumaCult™-ALI Medium Complete Kit (STEMCELL Technologies, Canada, Cat# 05001) including PneumaCult™-ALI Basal Medium-450mL, PneumaCult™-ALI 10X Supplement-50mL, and PneumaCult™-ALI 100X Maintenance Supplement-5mL and supplemented with: 0.02% Heparin Solution-1mL (STEMCELL Technologies, Cat# 07980), 0.48μg/mL Hydrocortisone (STEMCELL Technologies, Cat# 07925), 100U/mL Penicillin/Streptomycin (PenStrep) (Thermofisher, MA, Cat# 15140122), and 250μg/mL amphotericin B (AmpB) (Thermofisher, MA, Cat# 15290018) (Other bronchial epithelium maintenance mediums can likely be used in place of PneumaCult™-ALI Maintenance Medium but have not been tested. Regardless of medium used, sufficient antibiotics and antimycotics must be included to prevent contamination. Depending on animal model and culturing conditions, other antibiotic/antimycotic cocktails may be required to prevent contamination.)

## Sterile cassette

i. BioLite™ 60mm cell culture dish (Thermofisher, MA, Cat# 130181) or equivalent (Any sterile culture dish will suffice if it has a diameter large enough to hold the trachea and sufficient depth to hold enough medium to submerge the silicon insert and reach the Surgifoam™ without overflowing. A depth of 12.5mm is adequate. 6-well culture plates may also be used but may increase the risk of cross contamination.)

ii. Silicon rubber strip, 1/8 x 1 in, cut into 40mm segments

iii. Nontoxic/waterproof Original Gorilla Glue™ (Gorilla Glue Inc, OH) or nontoxic/waterproof equivalent (Other adhesives may be used in replace of Gorilla glue to anchor the silicon strip to the culture dish. When choosing an adhesive, it must be nontoxic and completely waterproof to avoid any undesired toxicity to the tissue to and to prevent the silicon from separating from the dish. The adhesive must completely dry and cure before use to prevent possibly toxicity form solidifying agents and to ensure a strong adherence.)

## Other

i. Surgifoam™ absorbable gelatin sponge (12x7mm) (Ethicon, NJ, Ref # 1972)

ii. BD™ Needle 1/2 in, 27g (BD, NJ, Ref# 305109) (6 needles (3 each side) are required to adequately pin the trachea and expose the entirety of the lumen from proximal to dorsal end. If imaging of the airway epithelium is not required, pinning the trachea may not be necessary, although trachea must maintain adequate contact with gelatin sponge for sufficient media exposure. 27g is the optimal size for pinning the trachea as it provides enough surface area to anchor into the silicon pad, but it does not cause any apparent damage to the tissue. Other types of pins, such as insect pins, should not be used, even if they are the same diameter as the needle. Unlike the needle, which easily slides through tissue, insect pins crush the tissue as they pass through leading to necrosis of the surrounding area.)

iii. Corning® 15 mL centrifuge tubes (Corning, AZ, Cat# CLS430791)

iv. Dulbecco's phosphate-buffered saline (DPBS) with calcium and magnesium (Thermofisher, MA, Cat# 14040133)

## Methods

### 1. Animal models

All animal experiments at UAB were conducted in accordance with UAB Institutional Animal Care and Use Committee (IACUC) approved protocols. All animal experiments used Sprague-Dawley CFTR$^{-/-}$ rats or their littermate wild-type (WT) controls as previously described [15]. Animals were bred and housed in standard cages with a 12-h light/dark cycle with ad libitum access to food and water and were routinely monitored. WT and CFTR$^{-/-}$ rats of the same sex were co-housed from time of weaning to study conclusion. Weaned rats were maintained on a standard rodent diet supplemented with DietGel 76A (Clear H2O, Westbrook, ME) and water containing 50% Go-LYTLEY (Braintree Laboratories, Inc., Braintree, MA) to reduce mortality from gastrointestinal obstruction [16]. Animals were euthanized by intraperitoneal injection of 500 µL pentobarbital sodium (390 mg/mL) followed by exsanguination of the hepatic portal vein. All animal procedures occurred after euthanasia at planned timepoints. Animals used in this study were ≥ 6 months to allow maturation of CF airway phenotype as described previously [16]. All experimental groups were matched by age and sex, with no noted sex differences in any outcomes.

### 2. Isolating and explanting *ex vivo* tracheae

**2.1 Preparing culture cassette.** Prepare culture cassettes at least 24 hours prior to use. Add a minimal drop of nontoxic and waterproof adhesive to the bottom of a 60mm culture dish and place a 25mm by 40mm silicon strip onto the glue, firmly pressing down until it lays evenly across the bottom of the dish. Allow the cassettes to cure in open air for at least 24 hours.

**2.2 Prior to explanting *ex vivo* trachea.** Prepare 27-gauge needles by detaching the needle shaft from its hub by bending the shaft back and forth with a pair of forceps. Store them in a 60mm dish until use. Prepare 6 to 8 needles for each trachea. Place Surgifoam™ sponges, within sterile packs, inside a culture hood, and cut them into segments of desired length (See 2.5). Saturate all culture cassettes, needles, forceps, scissors, and a petri dish in 70% ethanol and place them in culture hood. Once dry, expose all components to UV for 20 minutes.

**2.3 Harvesting and explanting *ex vivo* trachea.** Carefully expose the trachea from the larynx to the main branch point with special care not to damage or nick the trachea (Fig 1A). Cut the trachea directly below the larynx and above the main branch point, and then clean the trachea by removing fascia and any excess tissue. Immediately place the trachea in 5mL Pneumacult media containing PenStrep and AmpB on ice and leave for 1 hour (Incubating the freshly excised trachea in media containing an antibiotic/antimycotic cocktail on ice allows the trachea to equilibrate to an *ex vivo* environment in fully submerged media conditions. In addition, this is the first step that rinses the trachea and allows the antibiotic/antimycotic cocktail to begin antagonizing any microbes currently present on the trachea. This step can be augmented with longer incubation times and different cocktails if contamination becomes a recurrent issue.). After 1 hour, transfer the trachea to a sterile culture hood and place it into a petri dish for preparation. Identify the dorsal side of each trachea recognized by the side lacking continuous cartilage rings. Cut the length of the trachea from the distal to the proximal end on the dorsal side (Fig 1B). Place pre-cut Surgifoam™ sponges on top of the silicone strips inside a culture cassette (Fig 1C). Place trachea ventral side down onto the Surgifoam™

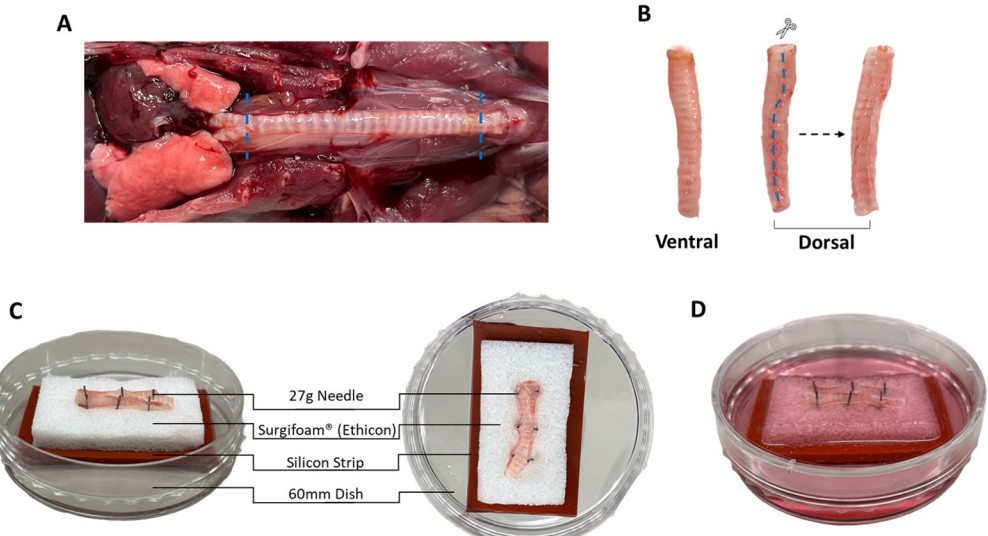

**Fig 1. Isolating and explanting *ex vivo* tracheae. (A)** Exposed trachea highlighting the points of excision above the main bronchial branch and below the larynx (blue dotted lines). **(B)** Excised and cleaned trachea highlighting where to cut on the dorsal face of the trachea from the distal to the proximal end to expose the lumen (blue dotted lines). **(C)** Top down and side views of a trachea pinned to a sterile culture cassette with labeled components showing how the trachea is pinned to the cassette. **(D)** Completed *ex vivo* trachea culture after media has saturated Surgifoam™ sponge.

sponge. Starting on the proximal end, pin three to four 27g needles through one outer wall of the trachea. Pass the needles through the Surgifoam™ and into the silicone strip (When pinning the trachea, it is important to pin it as close to the peripheral edge of the trachea as possible that still provides adequate surface area to grip the needles. This will increase the surface area of the trachea in contact with the gelatin sponge as well as prevent unnecessary damage to the trachea lumen. Furthermore, the pins must be inserted at an angle that will partially flatten the trachea to the sponge without causing tension on the trachea between the opposing sides. This is accomplished by pinning one side at a time, and by pinning straight down through the trachea after pushing the trachea flat with the needle.). Repeat this process on the opposite trachea wall. Rinse the trachea 3X with 5 mL of cold DPBS. Add cold Pneumacult media with PenStrep and AmpB to the cassette until the media reaches the Surgifoam™ (Fig 1D). Place a few drops of media on top of the Surgifoam™ surrounding the trachea to provide hydration while the sponge saturates (The Surgifoam™ sponge takes 1 to 2 hours to saturate after adding media. Placing drops on and around the trachea after initially pinning will provide the trachea with media during this time. The initial rinsing of the trachea should be performed before the sponge saturates, since the washes will not dilute any possible contaminants within the sponge after saturation.). Place the trachea into a sterile incubator at 37 ˚C with 5% $CO_2$. Rinse the trachea 3X with warm DPBS and replenish media daily for the following 2 days. After day 3, rinse the trachea 3X with DPBS and replenish media thrice weekly, indefinitely.

**2.4 Washing prior to experimentation.** Wash the trachea 24–72 hours prior to experimentation to remove any accumulated mucus from the trachea surface. To wash the trachea, pipette warm DPBS onto the surface of the trachea until covering the trachea by capillary action. Incubate the tracheae for 15 minutes and then aspirate DPBS. Add more DPBS and incubate the trachea for 45 minutes. Remove the DPBS. Additional washes can be performed if needed. When performing studies on mucus, ensure that mucus is adequately removed from the tracheae prior to experimentation. Although DPBS washes were sufficient to remove accumulated mucus in the following studies, washes with reducing agents such as 3mM

dithiothreitol, as described on HBEs, may need to be implemented if mucus remains after multiple DBPS washes [21].

**2.5 Treating *ex vivo* trachea with pharmacological agents.** Since the Surgifoam™ sponge retains fluid even after media removal from the culture cassette, special consideration must be taken when calculating concentrations of drugs used for experimentation to prevent undesired dilution of the drug. Prior to explanting the trachea, determine the optimal length of the Surgifoam™ sponge segment that will be used for culturing the trachea and cut a segment to this length. Saturate the sponge segment in DPBS and weigh it. Desiccate the sponge overnight and weigh it again. Calculate the difference between wet and dry weight in grams. Assuming the density of media is roughly 1 g/mL, the following equation can be used to calculate the drug concentration in the media that needs to be prepared to attain a desired target concentration once added to cassette with a pre-saturated Surgifoam™ sponge:

$$\left(\frac{(X + Y)}{X}\right) * Z = [Drug\ in\ Treatment\ Media]$$

X = Volume of treatment media to be added to cassette
Y = Difference in wet and dry weight of sponge in grams
Z = Target drug concentration

Although not performed in the following studies, further validation of actual drug concentrations in the media following this step may be necessary depending on the study being performed.

## 3. Histopathology of trachea

Cross sections of the medial trachea were immersion fixed in 10% neutral buffered formalin and submitted to histology laboratory for tissue processing, paraffin embedding and sectioning. Tissue sections were stained with alcian blue-periodic acid Schiff's (AB-PAS). Images were taken at 40X-200X magnification with a Nikon DS-Fi3 camera using a Nikon Eclipse Ts2 microscope (Tokyo, Japan). Goblet cells were quantified by counting the number of goblet cells over a given length of the trachea. Tracheae used for histopathology analysis were not used in any other analyses.

## 4. µOCT Imaging

Measurements of the functional microanatomy of *ex vivo* and freshly excised rat tracheae were performed using micro-optical coherence tomography (µOCT), a high-speed, high-resolution microscopic reflectance imaging modality [22]. The µOCT instrument provides cross-sectional images of the epithelium with a sub-micron resolution sufficient to directly visualize and quantify micro-anatomic parameters including air surface liquid (ASL) depth, periciliary liquid (PCL) depth, mucociliary transport (MCT) rates, ciliary beat frequency (CBF), and cilia coverage. The sub-micron capabilities of the µOCT provide the ability to measure these parameters free of any exogenous dyes or labels. Images are acquired at a rate of 40 frames per second and at 512 lines per frame. ASL and PCL depths were quantified directly by geometric measurement of the respective layers using ImageJ (NIH) software [23]. MCT rate was determined using time elapsed and distance traveled of native particles in the mucus layer over multiple frames. Ciliary beat frequency (CBF) and cilia coverage were investigated by Fourier analysis of the reflectance due to beating cilia using MATLAB. For consistency, the tracheae were placed in the same proximal to distal orientation and the imaging beam was placed at six standardized locations along the ventral surface of the trachea [16, 23].

## 5. Analysis of glands and 3D rendering using μOCT

Quantification of percent reflectivity within submucosal glands was achieved by measuring the area of reflectivity within a gland and dividing it by the total gland area. Single frames were used for analysis and areas were deemed reflective if they were above zero, which is the mean grey value of the background. All glands present within each μOCT video were used for quantification. Gland and ducts were evaluated over the whole length of the trachea. Three dimensional (3D) μOCT imaging was performed by collecting images with a 2000 μm amplitude across 300 frames. 3D reconstructions were rendered using ImageJ software [22].

## 6. PAS slot blot of secreted mucus

Equal volumes of samples in PBS were transferred onto a 0.45μm nitrocellulose membrane under gentle vacuum using Bio-Rad Bio-Dot SF microfiltration apparatus (Bio-Rad, Life Sciences, USA). Slot blots were removed from apparatus and subjected to periodic acid-Schiff's (PAS) reagent (Thermo Scientific, Grand Island, NY, USA) staining as described previously [24]. Blots were air dried overnight and imaged using HP color scanner. ImageJ software was used to perform densitometry measurements.

## 7. Statistics

All statistical analyses were performed in Graphpad Prism (GraphPad, CA, US) version 9.0 or later. Comparisons of cilia coverage and CBF were made using a two-way mixed effects ANOVA followed by a Tukey's multiple comparison. Comparisons between WT trachea and CF trachea at baseline and after carbachol stimulation were made using a two-way mixed effects ANOVA followed by Sidak's multiple comparison. All other statistics were performed using paired or unpaired t-tests as appropriate. A p-value of less than 0.05 was considered significant for all comparisons. All statistics are presented as mean ± SEM.

## Results

### *Ex vivo* tracheae remain viable for weeks after explanting

A key necessity of culturing tracheae *ex vivo* successfully is sustaining viability so that they maintain properties of a fresh trachea throughout experimentation on a consistent basis. To evaluate the viability of the *ex vivo* tracheae, tracheae were explanted per the aforementioned method (*see 2.3*) and fixed after 7 days to evaluate via histopathology. Fresh tracheae were harvested and processed for comparison. Cross sections of *ex vivo* and fresh tracheae were stained with AB-PAS to visualize the anatomy of the tracheae. Representative images at 40X and 200X show that the epithelium, lamina propria, submucosa, and other tissue of the *ex vivo* tracheae are intact and maintain normal morphology that is comparable to the fresh tracheae (Fig 2A and 2B). Furthermore, the epithelium of the *ex vivo* trachea is uniform with clearly defined ciliated and goblet cells, with cell distribution and morphology that was comparable to freshly fixed epithelium.

To evaluate the longevity of the *ex vivo* trachea viability, we implemented μOCT imaging to assess the degree of motile cilia expression (measured as percentage of active cilia coverage) and ciliary beating frequency as a metric of trachea functional status [22, 25]. Percent cilia coverage and CBF were used as indicators of normal epithelial function [26–28] and are non-invasive, allowing imaging of the same trachea repeatedly. An equal number of WT and CFTR$^{-/-}$ *ex vivo* tracheae were explanted, given 5 days to equilibrate, and imaged for cilia function analysis at day 5, 9, 14, and 21 to assess durability and stability. Fresh tracheae, with an equal number from WT and CFTR$^{-/-}$ rats, were harvested and imaged immediately upon explant for

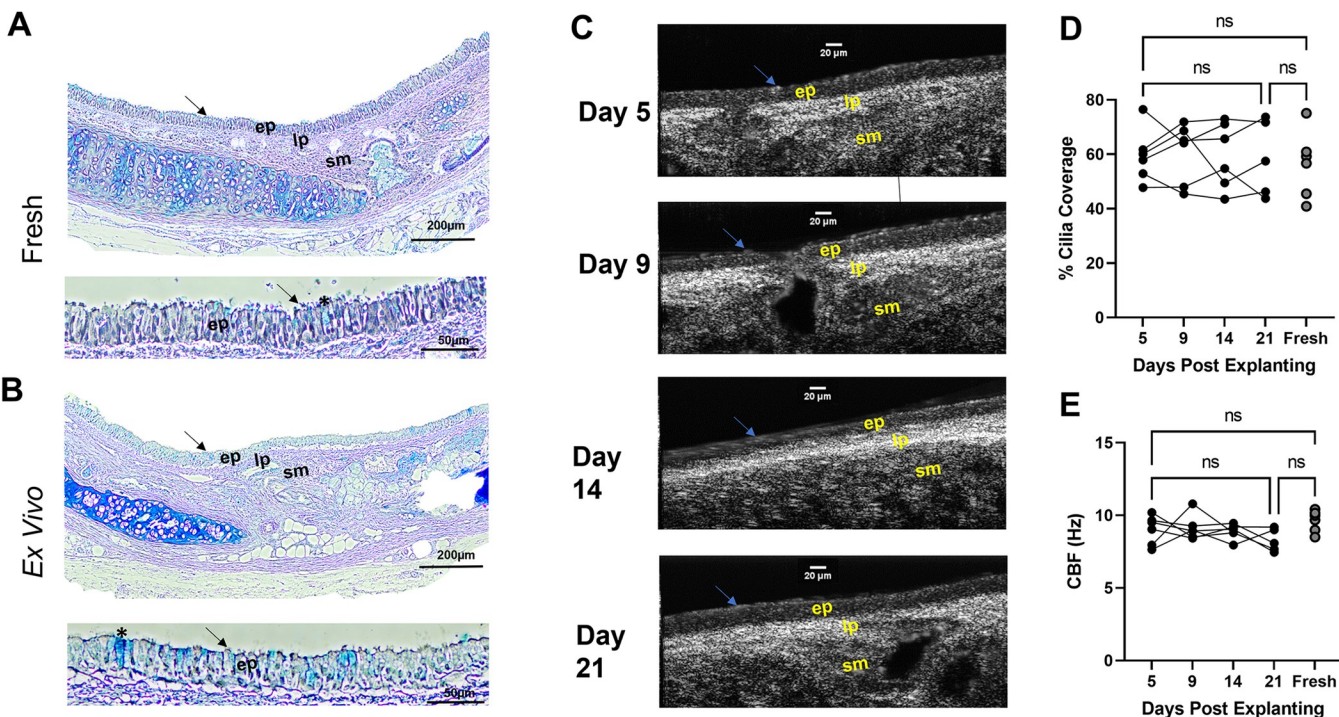

**Fig 2. *Ex vivo* tracheae remain viable for weeks after explanting. (A-B)** Representative AB-PAS stained tissue of **(A)** freshly excised trachea and **(B)** *ex vivo* trachea 7 days after explanting, depicting cilia (black arrow), goblet cells (*), epithelium (ep), lamina propria (lp), and submucosa (sm). Scale bars are 200 μm and 50 μm, respectively. **(C)** Representative μOCT images of *ex vivo* trachea 5, 9, 14, and 21 days after explanting depicting cilia (blue arrow), epithelium, lamina propria, and submucosa. Scale bars are 20 μm. Quantification of **(D)** % cilia coverage and **(E)** CBF from explanted trachea after 5, 9, 14, and 21 days and freshly excised trachea for comparison. N = 5-6/condition. nsP>0.05 by two-way mixed effects ANOVA followed by a Tukey's multiple comparison. AB-PAS-alcian blue-periodic acid Schiff's, μOCT- micro-Optical Coherence Tomography, CBF- ciliary beating frequency.

comparison, as previously described [16]. Representative averaged Z-projections, derived from imaging videos of the same trachea across all timepoints, showed that the epithelium, lamina propria, submucosa, and cilia remain intact over a 21-day span (S1–S4 Videos, Fig 2C). Quantification of the degree of cilia coverage across the epithelium (Fig 2D, S1 Fig) and CBF (Fig 2E) demonstrated that there was no meaningful quantitative difference in cilia coverage or CBF between freshly harvested trachea and *ex vivo* trachea at any timepoint. In addition, there was no meaningful decline in functional ciliary expression or CBF between *ex vivo* tracheae at five days compared to 21 days. This study was conducted through 21 days, but there was no indication that the trachea had declining activity at this time. Overall, these data suggest that the *ex vivo* trachea model remains viable and retains normal epithelial function for at least 21 days after explanting.

### *Ex vivo* tracheae exhibit epithelial and submucosal gland mucus dysfunction

It has been previously established that the $CFTR^{-/-}$ rat model exhibits depleted ASL and PCL shortly after birth and exhibits deficits in MCT by six months of age, as characterized by the analysis of freshly excised trachea [15, 16]. For the *ex vivo* model to be efficacious for CF mucus studies, it is critical that it retains the phenotypic characteristics in mucociliary physiology between diseased states after being cultured outside of its *in vivo* environment. To compare the functional microanatomy of the WT and $CFTR^{-/-}$ *ex vivo* trachea, WT and $CFTR^{-/-}$ tracheae were explanted and imaged after 7 days via μOCT. The tracheae were washed with

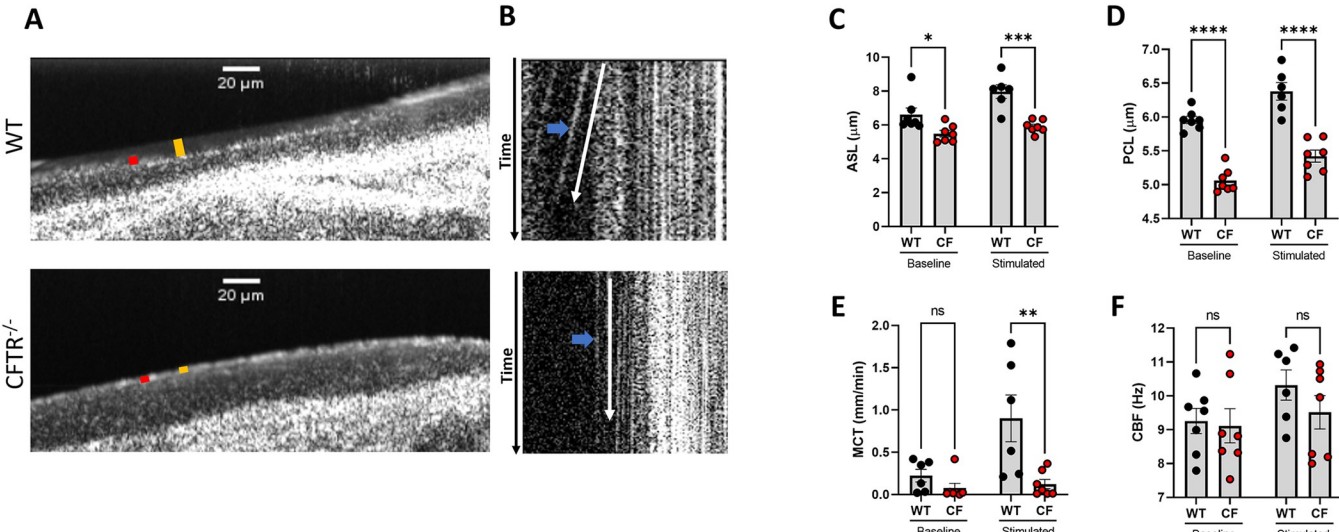

**Fig 3. *Ex vivo* tracheae exhibit epithelial mucus dysfunction. (A)** Representative μOCT images of WT and CFTR[-/-] *ex vivo* tracheae treated with carbachol to stimulate mucus secretion, depicting the ASL (orange bar) and PCL (red bar). **(B)** Time-dependent reprocessed images show tracks of mucus particles above the epithelial surface of WT and CFTR[-/-] *ex vivo* tracheae; the more horizontal direction of particle streaks (blue arrow) indicates more rapid transport. **(C-F)** Regions of interest were measured and averaged for each trachea for **(C)** ASL, **(D)** PCL, **(E)** MCT, and **(F)** CBF at baseline and after carbachol stimulation. N = 6-7/condition. nsP>0.05, *P<0.05, **P<0.01, ***P<0.001, ****P<0.0001 by two-way mixed effects ANOVA followed by a Sidak's multiple comparison. Scale bars are 20 μm. μOCT- micro-Optical Coherence Tomography, ASL- airway surface liquid, PCL-periciliary liquid, MCT- mucociliary transport, CBF-ciliary beating frequency.

the aforementioned protocol 72 hours prior to imaging to assure accumulated mucus did not contaminate functional assessments, and instead reflected steady state conditions. Videos were collected at baseline and after a 1-hour stimulation with 10uM carbachol, a cholinergic agonist and secretagogue, to observe under conditions where mucus transport was stimulated [12, 29]. Areas of mucus pooling that can occasionally occur in the proximal (i.e. caudal) area of the trachea were avoided. Representative averaged Z-projections, derived from imaging videos, showed depletion of ASL and PCL depths in CFTR[−/−] trachea compared to WT after stimulation with carbachol (Fig 3A). Quantitative analysis of each trachea showed ASL and PCL were significantly reduced in the CFTR[−/−] trachea at both baseline and after stimulation (ASL 5.9 ± 0.1 μm; PCL 5.4 ± 0.1 μm) compared to WT (ASL 8.0 ± 0.4 μm, P<0.001; PCL 6.4 ± 0.1 μm, P<0.0001; Fig 3C and 3D). Time-dependent image reprocessing showed a deficit in MCT of the CFTR[−/−] trachea at baseline compared to WT, although heterogeneity in CF was apparent and thus this comparison lacked statistical significance Fig 3E); however the difference in MCT after carbachol stimulation was exaggerated between CFTR[−/−] trachea (0.1 ± 0.1 mm/min) compared to WT (0.9 ± 0.7 mm/min; P<0.01; Fig 3B, 3E, S5 and S6 Videos), showing a significant deficit in MCT. There were no meaningful differences in CBF between WT and CFTR[−/−] trachea at baseline or after stimulation (Fig 3F), recapitulating previous findings on fresh trachea [15]. Histopathology of AB-PAS stained *ex vivo* trachea showed an increased number of goblet cells per length of epithelium in the CFTR[−/−] tracheae (0.015 ± 0.006 cells/mm) compared to WT (0.005 ± 0.002 cells/mm; P = 0.19; S2 Fig), although these differences were not statistically different, likely due to high variance in CFTR[−/−] tracheae.

CFTR[−/−] rats exhibit submucosal gland dysfunction by 6 months of age, including increased mucus gland hypertrophy and plugging of gland ducts [16, 30]. To determine whether the *ex vivo* trachea retained these features, we assessed trachea collected in Fig 3 for

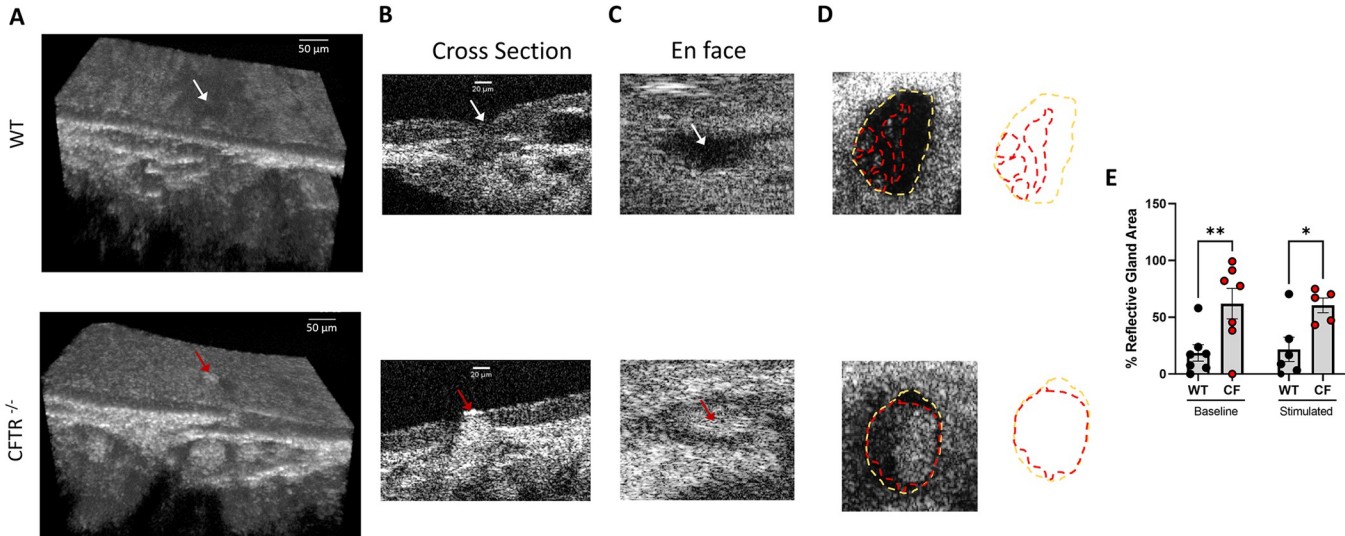

**Fig 4.** *Ex vivo* **tracheae exhibit submucosal gland mucus dysfunction.** (A) Three dimensional (3D) μOCT images of WT and CFTR^-/- *ex vivo* trachea representing a clear gland duct (white arrow) in WT and a plugged gland duct (red arrow) in CFTR^-/- trachea. (B) Cross sectional and (C) en face μOCT images of the same gland duct, depicting a clear duct (white arrow) and plugged duct (red arrow). (D) Representative μOCT cross sectional images of submucosal glands from WT and CFTR^-/- at baseline with tracings of the total gland area (yellow dashed line) and the reflective area indicative of mucus impaction (red dotted line). (E) Quantification of the reflective gland area over the total gland area for WT and CFTR^-/- trachea at baseline and after treatment with carbachol to stimulate mucus secretion. N = 5–7, *P<0.05, **P<0.01 by two-way mixed effects ANOVA followed by a Sidak's multiple comparison. Scale bars are 50 μm and 20 μm, respectively. μOCT- micro-Optical Coherence Tomography.

mucus impaction in glands and ducts via analysis of OCT videos. Representative 3D models of the trachea demonstrated substantial mucus within a gland and its associated gland duct in a CFTR^−/− trachea compared to a relatively clear gland and duct in WT trachea (S7 and S8 Videos, Fig 4A). Cross sectional and *en face* views of the CFTR^−/− gland ducts highlight an impacted duct of the CFTR^−/− trachea compared to a clear duct of the WT trachea (Fig 4B and 4C). To quantify this, we analyzed cross sections of glands for each trachea, as shown by the representative μOCT image demonstrating increased area of reflectivity within the gland of the CFTR^−/− tracheae compared to WT (Fig 4D). The reflective (i.e. opacified) area within the glands normalized for the total gland area show a significantly higher reflective gland area at baseline in the CFTR^−/− tracheae (60.6 ± 14.4% reflective gland area) compared to that of WT tracheae (21.7 ± 26.3% reflective gland area; P<0.05; Fig 4E); this effect persisted upon carbachol simulation (Fig 4E). Together, this data demonstrates that *ex vivo* CF trachea exhibits an aberrant CF airway phenotype at the level of the surface epithelium, consistent with previous reports of defects in the freshly excised trachea of CFTR^−/− rats. Furthermore, we observe novel evidence of CF airway gland defects, including mucus accumulation and stasis within the gland duct. Overall, these data support the use of ex vivo trachea analysis for studies on CF airway physiology.

## *Ex vivo* tracheae provide a tool for well-controlled pharmacological studies of airway physiology

Historically, *in vivo* pharmacological studies of the airway have been limited by the ability to dose airways under sufficient and constant concentrations of pharmacological agents for extended periods of time due to obstacles in airway delivery, such as the restrictive geometry of the lungs and rapid clearance by mucociliary transport [31]. Cell culture models, such as human bronchial epithelial cells (HBECs) circumvent many of these limitations but lack many

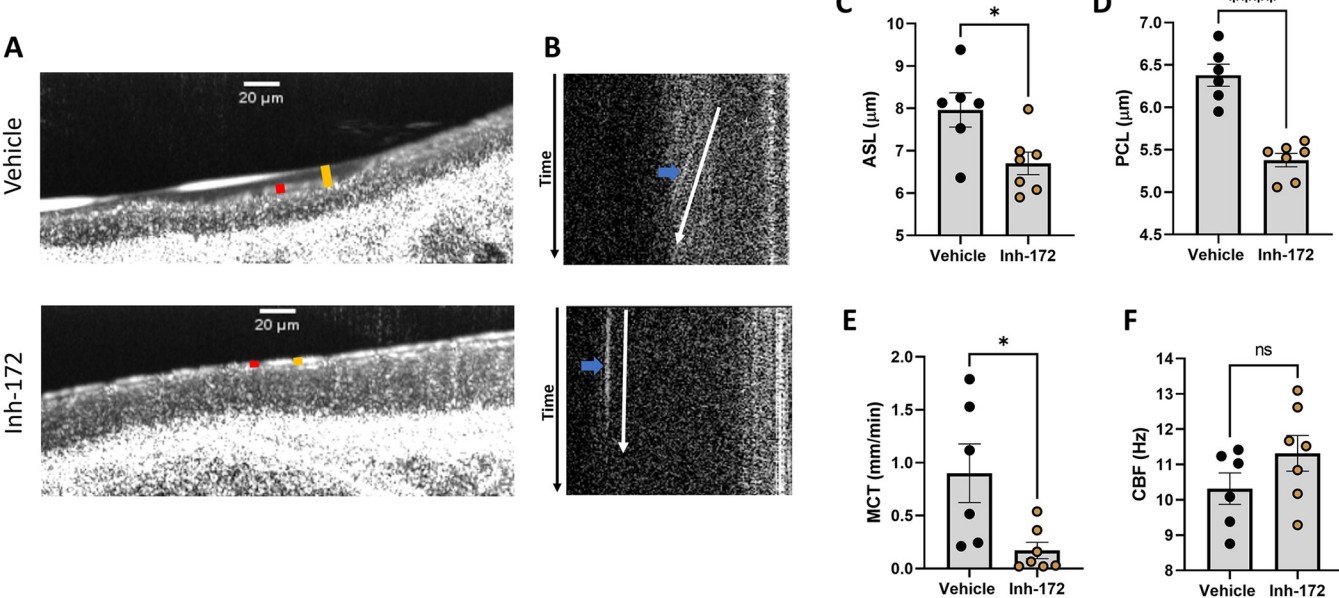

**Fig 5. *Ex vivo* tracheae provide a tool for well-controlled pharmacological studies of airway physiology. (A)** Representative µOCT images of WT *ex vivo* tracheae treated with 20 µM CFTR Inhibitor 172 (Inh-172) or vehicle control for 72 hours, depicting the ASL (orange bar) and PCL (red bar). **(B)** Time-dependent reprocessed images show tracks of mucus particles above the epithelial surface of WT *ex vivo* tracheae treated with Inh-172 or vehicle control; the more horizontal direction of particle streaks (blue arrow) indicates more rapid transport. **(C-F)** Regions of interest were measured and averaged for each trachea for **(C)** ASL, **(D)** PCL, **(E)** MCT, and **(F)** CBF. N = 6-7/condition. nsP>0.05, *P<0.05, ****P<0.0001 by unpaired t-test. Scale bars are 20 µm. µOCT-micro-Optical Coherence Tomography, CFTR- cystic fibrosis transmembrane conductance regulator, ASL- airway surface liquid, PCL- periciliary liquid, MCT- mucociliary transport, CBF- ciliary beating frequency.

*in vivo* features that are vital to accurately assess airway physiology, such as submucosal glands and complex tissues. An *ex vivo* trachea model provides the means to uniformly dose and assess an airway model that retains important features for studying airway physiology.

Previous work has shown that freshly excised tracheae treated acutely with cholinergic agonists consistently respond by increasing airway hydration as well as MCT [16, 32]. To determine if *ex vivo* trachea exhibit a normal response to acute treatments and may be useful for pharmacological studies, *ex vivo* tracheae were imaged via µOCT at baseline and after stimulation with 10 µM carbachol for 1-hour to the basolateral compartment. Representative µOCT images and quantifications (S3 Fig) show that ASL, PCL, and MCT were substantially increased after carbachol stimulation, recapitulating the responses previously reported after stimulation in fresh trachea [16, 32].

To determine whether the *ex vivo* model was useful for longer term studies, not possible in freshly excised tracheae, WT tracheae were explanted as described previously, and treated with 20 µM of the CFTR inhibitor 172 (Inh-172) or vehicle for 72 hours. Inh-172 is documented to reduce hydration and mucociliary transport within the airway by inhibiting CFTR-dependent anion transport [33–35]. After 72 hours, tracheae were imaged via µOCT to assess the functional microanatomy of the tracheae. Representative µOCT images shown in Fig 5A and quantification of ASL and PCL depth in Fig 5C–5D show that the ASL and PCL were significantly diminished after treatment with Inh-172 (ASL 6.7 ± 0.3 µm; PCL 5.4 ± 0.1 µm) compared to tracheae treated with vehicle (ASL 8.0 ± 0.4 µm, P<0.05; PCL 6.4 ± 0.1 µm, P<0.0001). In addition, time-dependent image reprocessing and quantification showed that WT trachea treated with Inh-172 exhibited significantly slower transport of particles within the mucus layer (0.2 ± 0.1 mm/min) compared to vehicle treated trachea (0.9 ± 0.3 mm/min; P<0.05;

S9 and S10 Videos, Fig 5B–5E). There were no meaningful differences in CBF between WT and CF trachea at baseline or after stimulation (Fig 5F). These data indicate that *ex vivo* trachea respond to longer-term pharmacological treatments not able to be tested on freshly excised trachea due to limited viability and may be useful for well controlled pharmacological studies where drug delivery is not in question.

## *Ex vivo* tracheae increase yields of secreted mucus

Studies involving secreted mucus from *in vivo* models are severely limited by the poor yield of mucus that can be collected from the airways. Yields from in vivo models can be increased by collecting form the whole bronchial tree, but this requires sacrifice of the animal and cannot be repeated. The unique properties of the *ex vivo* trachea model may enable it to serve as a continuous source for secreted mucus given that mucus accumulates on the proximal portion of the trachea, and the resulting substrate can be repeatedly collected from the same trachea over time. Photographs of a fresh and *ex vivo* trachea (Fig 6A) and cross sectional μOCT images of the proximal ends of those tracheae (Fig 6B) show that *ex vivo* trachea exhibit substantially increased pooling of mucus compared to freshly excised trachea. To compare the mucus yield quantitatively between *ex vivo* and freshly excised trachea, mucus was collected from individual *ex vivo* tracheae 72 hours after the wash maneuver; for comparison, mucus was also collected from individual, fresh tracheae following a 30-minute incubation after harvest. Individual mucus collections were solubilized in equal volumes of PBS and then subjected to a Bradford protein assay to compare the total secreted mucus protein yield per trachea. Quantifications of the Bradford assay show that significantly more protein was obtained from the *ex vivo* tracheae (103.6 ± 18.5 μg per trachea) compared to freshly harvested tracheae

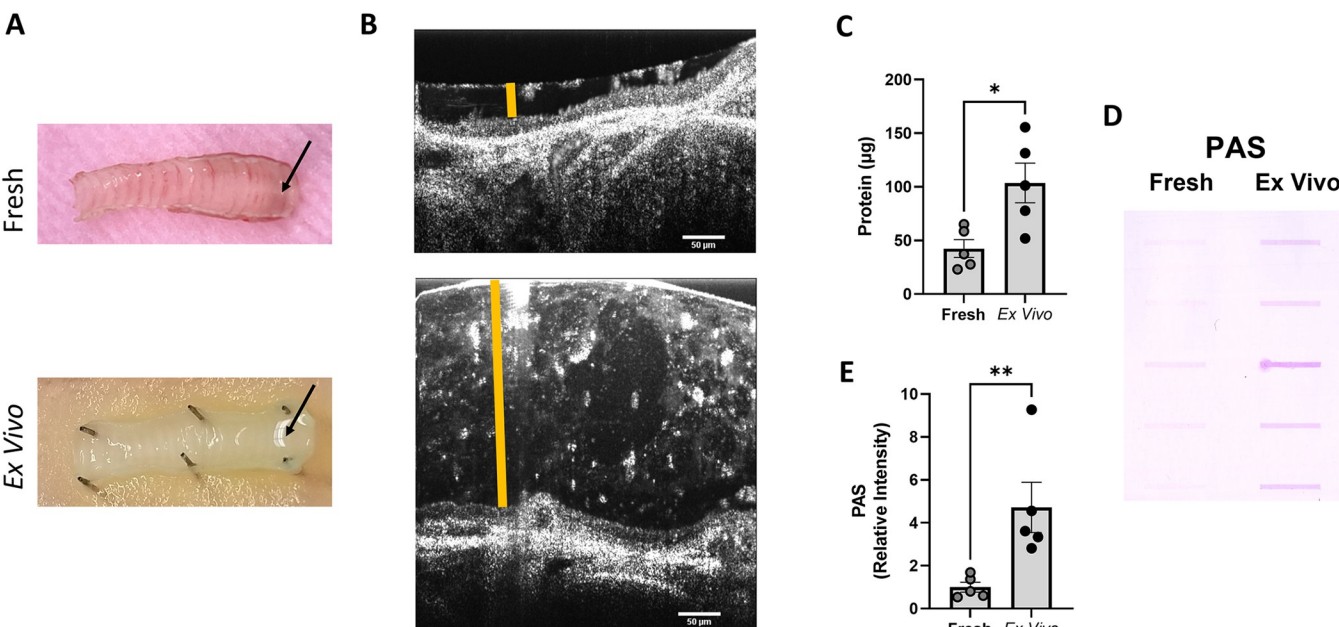

**Fig 6. *Ex vivo* tracheae increase yields of secreted mucus.** (A) Representative raw images of a fresh and *ex vivo* trachea, depicting mucus at the proximal end of the tracheae (black arrows). (B) Representative μOCT images of the proximal end of a fresh and *ex vivo* tracheae, depicting the ASL (orange bar). (C) Quantification of the total protein yield of secreted mucus collections from individual fresh tracheae or *ex vivo* tracheae 72 hours after washing. (D) Representative PAS slot blot of secreted mucus collections from individual fresh tracheae or *ex vivo* tracheae 72 hours after washing. Each slot represents a collection from a different trachea. (E) Quantification of PAS blot by densitometry. N = 5/condition. *P<0.05, **P<0.01 by unpaired t-test. Scale bars are 50 μm. μOCT- micro-Optical Coherence Tomography, ASL- airway surface liquid, PAS- Periodic acid- Schiff's.

(42.4 ± 8.3 μg per trachea; P<0.05; Fig 6C), indicating increase volume obtained was not due to diluted protein. In addition, equal volumes of mucus in PBS were subjected to a slot blot and stained for PAS, a common stain used to quantify mucin abundance [36]. A representative image (Fig 6D) and densitometry of the PAS blot revealed that there was significantly more PAS positive material collected from the *ex vivo* tracheae (4.7 ± 1.2 relative intensity) compared to the freshly harvested tracheae (1.0 ± 0.2 relative intensity; P<0.01; Fig 6E) despite equal volumes of mucus subjected to the assay. This data suggests that *ex vivo* trachea provide higher mucin protein yields from a single collection than freshly harvested tracheae and may serve as a readily available resource for mucus in studies requiring greater amounts of mucin than current *in vivo* models provide.

## Discussion

CF is a muco-obstructive disease hallmarked by dysfunctional airway microanatomy and aberrant mucus physiology [2, 3]. Current animal models used to study airway and mucus dysfunction in CF, such as the CF pig and CFTR$^{-/-}$ ferret, are costly and difficult to maintain [4, 9]. More recently, the CFTR$^{-/-}$ rat model has provided a means to study CF animals with aberrant mucus physiology in a more accessible model of CF [15, 16]. Although this model has been effectively used to study the airway microanatomy and mucus physiology, there remains a need for accessible models that permit well-controlled and longitudinal studies on airway physiology. Here we developed and characterized a novel *ex vivo* model, which retains CF pathology and responded to pharmacologic stimuli, providing a tool to perform consistent, longitudinal pharmacological studies on airway dysfunction in CF.

Previously, studies on excised trachea have been performed acutely, where the trachea is excised, equilibrated on media for a short time, and then used for experimentation due to short term viability [12, 16]. Although freshly excised tracheae have been shown to maintain *in vivo* airway features and normal function for short periods after excision, they ultimately become necrotic and contaminated without proper stabilization and culturing conditions. Furthermore, the airway physiology of fresh trachea may be altered during the period of study resulting from the invasive nature and mechanical stress of harvesting and preparing the trachea [37, 38]. A key feature of the *ex vivo* trachea model is that it retains the physiology of the donated tissue and remains without bacterial contamination and viable for at least 21 days after explanting. We further show that the epithelium, lamina propria, submucosa, and other tissue of the *ex vivo* trachea remain intact with normal morphology after 7 days post explanting compared to fresh trachea stained with AB-PAS (Fig 2A and 2B). In addition, these features remain intact and *ex vivo* tracheae maintain normal cilia coverage and CBF over a 21-day span (Fig 2C–2E, S1 Fig). This 21-day assessment not only shows viability but also highlights the application of this model for non-invasive longitudinal studies. The stabilized culture model and exposed trachea lumen allow repeated measures of the same tissue for a prolonged period under equilibrated conditions rather than a single measure from freshly excised, stressed tissue. This benefit may be utilized for studies requiring repeated measures of airway physiology, which are difficult or impossible to perform in current *in vivo* models.

Current models of CF airway pathology exhibit specific, aberrant airway and mucus features that distinguish them from healthy tissue. More specifically, the CFTR$^{-/-}$ rat model exhibits impaired mucociliary clearance, a depleted ASL and PCL, and submucosal gland plugging compared to its healthy counterpart [15, 16]. These phenotypic differences are vital for understanding the pathology of CF and pursuing therapies intended to address them. For the *ex vivo* model to be applicable to disease studies, it must retain the key characteristics of the disease such as CF. We show that the *ex vivo* trachea model retains significant deficits in ASL,

PCL, and MCT in CFTR$^{-/-}$ tracheae compared to WT tracheae (Fig 3A–3F), recapitulating previous findings that used freshly excised tracheae [15, 16]. This includes a depleted ASL, PCL, and impaired MCT of CFTR$^{-/-}$ tracheae compared to WT tracheae in both fresh and ex vivo tracheae. Despite these consistencies, we notice small differences. The overall ASL of the *ex vivo* trachea in both WT and CFTR$^{-/-}$is smaller than previously reported in freshly excised tracheae [16] (Fig 3A, 3C). This is most likely due to differences in the *ex vivo* environment in addition to the lack of ASL contribution from bronchi and distal airways that is present *in vivo*. The *ex vivo* model is also void of mechanical stresses that may have exaggerated the ASL of freshly excised tracheae during harvest [37, 38]. Having noted these small differences, the increased precision and capacity for measurements maintain discriminative capacity between CF and controls with reasonable number of tracheae.

We show that the *ex vivo* trachea exhibit submucosal gland plugging in the CFTR$^{-/-}$ trachea that is absent in WT tracheae (Fig 4A–4C), a novel observation possible due to the more precise morphometry and imaging that can be conducted on stabilized ex vivo tissue. Quantification of the reflective area within each gland compared to the total gland area, as a measure of mucus impaction in the glands, revealed that *ex vivo* CF trachea had substantially more mucus within submucosal glands compared to WT trachea, indicative of gland plugging (Fig 4D and 4E). These findings complement important mucus secretion studies conducted on porcine airways, which showed increased mucus in submucosal gland ducts and impaired mucus detachment from ducts in CF pigs compared to WT [12, 39]. Our *ex vivo* model can be utilized to further study this defective mucus release from glands in CF, since it exhibits CF submucosal gland duct disfunction and provides the means to longitudinally assess gland physiology. Overall, the retention of a CF airway phenotype in the *ex vivo* model provides evidence that this model may be useful for elucidating disease specific mechanisms in CF and potentially other diseases, as well as for testing therapeutics.

*In vivo* drug administration plays an important role in pharmacologic testing but has important limitations that can alter their translational potential. For treatments of the airway, administration to rats and other animal models is not as precise or controlled compared to the well-controlled environment of cell culture studies. Dosing can be limited or heterogenous with methods such as intravenous or intratracheal administration due to off target effects or inconsistent/insufficient delivery since the agents must cross the mucus barrier and are cleared by mucociliary transport at unknown rates [31, 40, 41]. In contrast, cell culture models are missing many *in vivo* features required to accurately assess airway physiology, such as submucosal glands and complex tissues. The model demonstrated here provides a tool to study physiological responses of the airway with precise and well-controlled pharmacological treatments in a model with many in vivo features intact.

To study application of pharmacological agents to the ex vivo trachea model, we confirmed that the *ex vivo* model responds to basolateral cholinergic stimulation as previously confirmed in freshly excised trachea [12, 16] (S3 Fig). Extending the models utility to longer term pharmacological studies, we utilized a well characterized agent, CFTR Inh-172, with known effects on CFTR-dependent ion transport. We show that a 72-hour treatment with Inh-172 depletes the ASL and PCL and impairs MCT on *ex vivo* trachea (Fig 5A), showing the CFTR dependence of these physiological effects. This study highlights the utility of the *ex vivo* model for mechanistic or therapeutic pharmacological studies that require consistent and accurate dosing, similarly to traditional cell culture models. Furthermore, many pharmacologic studies must first titrate an effective dose, requiring months of effort and a substantial number of animals. Since the *ex vivo* model can be accurately dosed under constant concentrations, it provides a tool to titrate pharmacological dosing accurately and quickly while saving time and minimizing distress to animals.

Historically, mucus has been collected from *in vivo* models via bronchoalveolar lavage, lung homogenate, or from freshly excised tissue [42]. These methods may produce inadequate yields or purity for many applications, including studies of mucin structure [43, 44]. Unlike tracheal mucus *in vivo*, *ex vivo* mucus is not swallowed or expectorated. Using raw images of tracheae in addition to μOCT cross sectional images of the proximal tracheae, we show that unlike fresh tracheae, the secreted mucus from *ex vivo* tracheae pools at the proximal end (Fig 6A and 6B), where it accumulates until collected, providing a tool to substantially increase mucus yields for many applications. Quantifications of secreted mucus collections using the Bradford assay show that *ex vivo* trachea provide significantly more secreted mucus protein than fresh trachea from a single collection (Fig 6C). Using PAS, we show that *ex vivo* tracheae provide roughly 4.7X the amount of mucin, the primary constituent of mucus, compared to fresh tracheae (Fig 6D and 6E), ruling out the possibility that the increase in pooled ASL and mucus protein from *ex vivo* tracheae is an artifact of an accumulation of fluid or cellular debris. Although the mucus yield from a single *ex vivo* collection is already higher than a single collection from fresh tracheae, likely due to mucus accumulation that would normally be cleared in an *in vivo* setting, the *ex vivo* mucus yield may be substantially increased by repeated collections on the same trachea, a practice not possible on freshly excised tissue. These data provide strong evidence that the *ex vivo* trachea model increases the yield of secreted mucus and may serve as a valuable resource for studies requiring higher mucus yields than *in vivo* models provide and help address an emerging limitation in acquisition of CF mucus in humans that is also contaminated by a variety of pathogens.

The *ex vivo* tracheae model is limited due to the sampling of only the tracheae, which excludes other regions of the lung that may be important for studying lung outcomes in CF or other diseases; however, the trachea has been widely utilized as an effective indicator of airway physiology outcomes in multiple disease states and animal models, due to its properties of fluid secretion and mucus production in glands and epithelia [12, 16, 45]. A comparable *ex vivo* model, precision cut lung slices (PCLS), has emerged as a versatile tool to study lung tissue function and outcomes within the context of native cell populations and resident immune cells across multiple regions of the lung. Although a valuable tool, PCLS are limited in their utility to evaluate lung physiology at ALI, particularly mucociliary physiology in the context of a larger tissue, due to their fixation by agarose inflation and sampling of a small tissue area with inherent inconsistencies of native cell populations due to the extensive heterogeneity of the lung [46]. The use of PCLS in conjunction with *ex vivo* tissue cultures may provide a powerful combination to comprehensively study respiratory responses in specific tissues as well as at the ALI. Another limitation to consider when using the *ex vivo* trachea model is the absence of non-resident immune cells, unless added extrinsically to the cultures. This may be an important consideration when studying airway physiology outcomes that may rely on the presence of immune cells. Additionally, although we show that cholinergic stimulation remains intact after 7 days post explant (S3 Fig), there will likely be a loss of extrinsic neural innervation, which has been shown to be important for epithelial and submucosal gland function [47].

In summary, we developed and characterized a novel CF rat *ex vivo* trachea model to study airway physiology and mucus in CF. The ability to culture tracheae *ex vivo*, while maintaining their viability, disease phenotype, and treatment responses may circumvent many limitations of current *in vivo* models. Although the CFTR$^{-/-}$ rat model was chosen for this study, these techniques may be extended to diseases, and animal models beyond CF, providing an effective tool to study airway pathogenesis in a multitude of models and disease states.

## Supporting information

**S1 Fig. Cilia coverage map of *ex vivo* trachea. (A)** Representative μOCT images with color map of cilia coverage on an *ex vivo* trachea at 5, 9, 14, and 21 days after explanting. Scale bars are 20 μm. μOCT- micro-Optical Coherence Tomography.
(TIF)

**S2 Fig. Goblet cells in WT and CFTR$^{-/-}$ *ex vivo* tracheae. (A)** Representative AB-PAS stained tissue of WT and CFTR$^{-/-}$ *ex vivo* tracheae 7 days after explanting, depicting goblet cells (*).
**(B)** Quantification of goblet cells per mm of tracheal epithelium. N = 3/condition. nsP>0.05 by unpaired t-test. Scale bars are 50 μm. AB-PAS- alcian blue-periodic acid Schiff's.
(TIF)

**S3 Fig. *Ex vivo* trachea respond to acute cholinergic stimulation. (A)** Representative μOCT images of WT *ex vivo* tracheae at baseline and after stimulation with 10 μM carbachol for 1 hour, depicting the ASL (orange bar) and PCL (red bar). **(B)** Time-dependent reprocessed images show tracks of mucus particles above the epithelial surface of WT *ex vivo* tracheae at baseline and after stimulation with carbachol; the more horizontal direction of particle streaks (blue arrow) indicates more rapid transport. **(C-F)** Regions of interest were measured and averaged for each trachea for **(C)** ASL, **(D)** PCL, **(E)** MCT, and **(F)** CBF. N = 5-7/condition. nsP>0.05, **P<0.01 by paired t-test. Scale bars are 20 μm. μOCT- micro-Optical Coherence Tomography, ASL- airway surface liquid, PCL- periciliary liquid, MCT- mucociliary transport, CBF- ciliary beating frequency.
(TIF)

**S1 Video. Representative μOCT video of trachea 5 days post explantation.**
(AVI)

**S2 Video. Representative μOCT video of trachea 9 days post explantation.**
(AVI)

**S3 Video. Representative μOCT video of trachea 14 days post explantation.**
(AVI)

**S4 Video. Representative μOCT video of trachea 12 days post explantation.**
(AVI)

**S5 Video. Representative μOCT video of wild type trachea *ex vivo* 1 hour post-carbachol stimulation.**
(AVI)

**S6 Video. Representative μOCT video of CFTR$^{-/-}$ trachea *ex vivo* 1 hour post-carbachol stimulation.**
(AVI)

**S7 Video. Representative μOCT three-dimensional reconstruction of wild type *ex vivo* trachea near an airway gland.**
(AVI)

**S8 Video. Representative μOCT three-dimensional reconstruction of CF *ex vivo* trachea near an airway gland.**
(AVI)

**S9 Video. Representative µOCT video of wild type trachea ex vivo treated with vehicle for 72 hours.**
(AVI)

**S10 Video. Representative µOCT video of wild type trachea ex vivo treated with CFTR$_{Inh}$-172 for 72 hours.**
(AVI)

**S1 Raw images.**
(PDF)

## Acknowledgments

We would like to thank Dr. Susan E Birket, PharmD, PhD and the CFTR Rat Models Core for providing all rats used in these studies. We would also like to thank UAB Comparative Pathology for processing samples for histopathology.

## Author Contributions

**Conceptualization:** Elex Harris, Jarrod Barnes, Steven M. Rowe.

**Data curation:** Elex Harris, Molly Easter.

**Formal analysis:** Elex Harris, Janna Ren.

**Funding acquisition:** Elex Harris, Jarrod Barnes, Steven M. Rowe.

**Investigation:** Elex Harris.

**Methodology:** Elex Harris, Stefanie Krick, Steven M. Rowe.

**Project administration:** Jarrod Barnes, Steven M. Rowe.

**Resources:** Jarrod Barnes, Steven M. Rowe.

**Supervision:** Stefanie Krick, Steven M. Rowe.

**Writing – original draft:** Elex Harris, Jarrod Barnes, Steven M. Rowe.

**Writing – review & editing:** Elex Harris, Molly Easter, Stefanie Krick, Jarrod Barnes, Steven M. Rowe.

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
