## [Decision Letter · Decision Letter 0]

5 Jul 2023

PONE-D-23-16864An ex vivo rat trachea model reveals abnormal airway physiology and a gland secretion defect in cystic fibrosisPLOS ONE

Dear Dr. Rowe,

Thank you for submitting your manuscript to PLOS ONE. After careful consideration, we feel that it has merit but does not fully meet PLOS ONE’s publication criteria as it currently stands. Therefore, we invite you to submit a revised version of the manuscript that addresses the points raised during the review process.

We look forward to receiving your revised manuscript.

Kind regards,

Abdelwahab Omri, Pharm B, Ph.D, Laurentian University, Canada

Academic Editor

PLOS ONE

Reviewers' comments:

Reviewer's Responses to Questions

**Comments to the Author**

1. Is the manuscript technically sound, and do the data support the conclusions?

Reviewer #1: Yes

Reviewer #2: Yes

Reviewer #3: Yes

2. Has the statistical analysis been performed appropriately and rigorously? 

Reviewer #1: Yes

Reviewer #2: Yes

Reviewer #3: Yes

3. Have the authors made all data underlying the findings in their manuscript fully available?

Reviewer #1: Yes

Reviewer #2: Yes

Reviewer #3: Yes

4. Is the manuscript presented in an intelligible fashion and written in standard English?

Reviewer #1: Yes

Reviewer #2: Yes

Reviewer #3: Yes

5. Review Comments to the Author

Reviewer #1: The manuscript “An ex vivo rat trachea model reveals abnormal airway physiology and a gland secretion defect in cystic fibrosis” describes development of an in vitro method to maintain excised tracheas for at least three weeks. This enables access to the tissue for pharmacological manipulation for longer times than fresh tissue while maintaining the cystic fibrosis phenotype of CFTR-/- rat tracheas. The major advantage with this approach compared to traditional ALI cultures is that this preparation includes submucosal glands. This enables the study of submucosal gland derived bundled strands and how they contribute to mucociliary clearance. As stated in the title, the preparation can also be used to study how mucus is altered in cystic fibrosis, where the bundled strands contribute to the pathology. The results presented are of great interest not only for scientists studying cystic fibrosis, but also other fields where the tracheal epithelium is studied. The results are in context of previous literature and the references are fair. The conclusions are supported by the results and the methods are described in enough detail for reproduction. The manuscript is very clear in organization and language. There are only some minor concerns which need to be addressed:

1: In line 287, it says: however the difference in MCT between CFTR−/− trachea (0.1 ± 0.1 mm/min) compared to WT (0.9 ± 0.7 mm/min; P<0.01; Figure 3E) was exaggerated after carbachol stimulation (Videos 5-6, Figure 3B,E). Quantification of MCT for each trachea showed a significant deficit in MCT after stimulation in CFTR−/− trachea (0.1 ± 0.1 mm/min) compared to WT (0.9 ± 0.7 mm/min; P<0.01; Figure 3E).

The numbers given are the same for unstimulated and stimulated tracheas, but in the figure it is clear that the numbers are different. There must be a mistake somewhere.

2: In line 513, the text reads: Using PAS, we show that ex vivo tracheae provide roughly 4.7X the amount of mucin, the primary constituent of mucus, compared to fresh tracheae (Figure 6D-E), ruling out the possibility that the increase in pooled ASL and mucus protein from ex vivo tracheae is an artifact of an accumulation of fluid or cellular debris.

This conclusion is most likely correct, but there is no explanation given for why there would be more mucus in the cultured tracheas. Was the number of goblet cells increased during culture? Was the number of mucus-producing cells in the glands increased? Is the explanation instead that mucus is collecting on the cultured trachea that would be removed by mucociliary clearance in the live animal? This deserves some further discussion.

Reviewer #2: Harris et al. have set out a rat trachea model to study the abnormalities of the CF respiratory tract ex vivo. The paper describes a convincing, very carefully detailed method to isolate and maintain tracheae for ex vivo experimentation and drug testing.

Main concerns:

1. Methods, Lines 180-185: This part can sound a bit confusing for someone who is not well acquainted with the topic. Have the whole tracheae undergone fixation and paraffinization for histopathology or just a part of each trachea? Also, step 4 says that µOCT was performed in freshly excised tracheae, which, I suppose, had not yet been processed for histological analysis. Kindly clarify the order in which these analyses – histopathology and imaging – were performed.

2. Results: When downloaded, some figures have dark backgrounds, which make them difficult to see. Please make sure that all figures are visible in case the paper is sent for another review round.

3. Lines 293-295: The word “trend” does not sound as the most appropriate. In general, trend is used to describe a change over time or, if much, a pattern after repeated measurements with different sample. In the present study, only a few animals have been used, which is in accordance with ethical protocols, so the amount may not be representative enough to point out any trend.

4. Figure 2: Were the experiments in Figure 2 conducted with CFTR-/- tissue or with WT tissue? If conducted with WT tissue, how long were the CFTR-/- tracheae viable?

5. Discussion: Please elaborate on the added value, difference and importance of this model compared to precision cut lung slices.

6. Please elaborate (discussion) on the contribution of trachea impairment in CF compared to the rest of the lung.

Minor concerns:

1. This is merely technicality, but, overall, the authors mix the Results section with data interpretation, which, in turn, should be in the Discussion section. I would advise the authors to order the results in terms of what the authors consider most important, only with the description of the results. Likewise, the authors can interpret these findings in the Discussion section by following the same order in the result. This can make it easier to the reader to follow the authors’ train of thought.

2. Please avoid repetitive statements in the M&M (e.g., first sentence of 2.1 and 2.2).

3. The quality of the representative image of Figure S2 of the WT animal is not comparable to the CFTR-/- animal and should be replaced.

4. Include full names of abbreviations at the end of every figure legend as figure should be able to stand alone without reading the main text.

5. Please specify whether the ducts analyzed for Figure 4 were analyzed in the same trachea region (proximal vs. distal).

6. Line 325: small typing error “previously” should read “previous”

7. For Figure 4: please provide a supplemental figure showing clear ducts in both WT and CFTR-/- tracheae after the washing protocol.

Reviewer #3: Introduction

Lines 60- 61, page 9 – the statement “and not easily amenable to genetic modification or longitudinal studies” should be softened. There are several genetically modified pigs and ferrets available. The demand for genetically modified models in these species may be less, but that doesn’t mean they are not amenable to genetic modification. Similarly, there several longitudinal studies in ferrets and pigs. It is recommended that this statement be softened or re-worded.

Methods

2.4 – is the washing procedure adequate to remove mucus from the CFTR-/- rat airway? For example, mucus is likely more adherent in CFTR-/- rat airways.

2.5 – has the equation listed been validated? Meaning, have the expected concentrations of drugs using that equation been measured in the media and/or in the tissues?

6.0 PAS Slot Blot – Is it assumed that PAS is only detecting secreted mucins and not other glycoproteins or glycolipids? Additionally, PAS since only detects neutral mucins, are conclusions draw about secreted mucins accurate (since some literature suggests that mucins in CF may be more acidic in form, and would be detected by Alcian blue preferentially.

Results

Were sex differences noted?

The images in Figure 2 for ex vivo trachea give impression there is more vacuoles in the trachea – is that a true observation or just due to the image selected?

Discussion

Nicely written. A recommendation is to include a limitations paragraph. For example, one limitation is that ex vivo preps do not allow for studying the interaction of the epithelia with the immune cells, unless immune cells are added. Additionally, though cholinergic responses remained intact, there will be a loss of extrinsic neural innervation, which is important for surface epithelia and submucosal gland function.

6. PLOS authors have the option to publish the peer review history of their article (what does this mean?). If published, this will include your full peer review and any attached files.

Reviewer #1: No

Reviewer #2: No

Reviewer #3: No

---

## [Author Response · Author response to Decision Letter 0]

22 Sep 2023

Reviewer #1: The manuscript “An ex vivo rat trachea model reveals abnormal airway physiology and a gland secretion defect in cystic fibrosis” describes development of an in vitro method to maintain excised tracheas for at least three weeks. This enables access to the tissue for pharmacological manipulation for longer times than fresh tissue while maintaining the cystic fibrosis phenotype of CFTR-/- rat tracheas. The major advantage with this approach compared to traditional ALI cultures is that this preparation includes submucosal glands. This enables the study of submucosal gland derived bundled strands and how they contribute to mucociliary clearance. As stated in the title, the preparation can also be used to study how mucus is altered in cystic fibrosis, where the bundled strands contribute to the pathology. The results presented are of great interest not only for scientists studying cystic fibrosis, but also other fields where the tracheal epithelium is studied. The results are in context of previous literature and the references are fair. The conclusions are supported by the results and the methods are described in enough detail for reproduction. The manuscript is very clear in organization and language. There are only some minor concerns which need to be addressed:

1: In line 287, it says: however the difference in MCT between CFTR−/− trachea (0.1 ± 0.1 mm/min) compared to WT (0.9 ± 0.7 mm/min; P<0.01; Figure 3E) was exaggerated after carbachol stimulation (Videos 5-6, Figure 3B,E). Quantification of MCT for each trachea showed a significant deficit in MCT after stimulation in CFTR−/− trachea (0.1 ± 0.1 mm/min) compared to WT (0.9 ± 0.7 mm/min; P<0.01; Figure 3E).

The numbers given are the same for unstimulated and stimulated tracheas, but in the figure it is clear that the numbers are different. There must be a mistake somewhere.

R1. Thank you for catching this error that occurred during the revision process. This has been corrected as follows.

Altered Sentence: Time-dependent image reprocessing showed a deficit in MCT of the CFTR−/− trachea at baseline compared to WT, although heterogeneity in CF was apparent and thus this comparison lacked statistical significance Figure 3E); however the difference in MCT after carbachol stimulation was exaggerated between CFTR−/− trachea (0.1 ± 0.1 mm/min) compared to WT (0.9 ± 0.7 mm/min; P<0.01; Figure 3B,E, Videos 5-6), showing a significant deficit in MCT.

Removed Sentence: Quantification of MCT for each trachea showed a significant deficit in MCT after stimulation in CFTR−/− trachea (0.1 ± 0.1 mm/min) compared to WT (0.9 ± 0.7 mm/min; P<0.01; Figure 3E).

2: In line 513, the text reads: Using PAS, we show that ex vivo tracheae provide roughly 4.7X the amount of mucin, the primary constituent of mucus, compared to fresh tracheae (Figure 6D-E), ruling out the possibility that the increase in pooled ASL and mucus protein from ex vivo tracheae is an artifact of an accumulation of fluid or cellular debris.

This conclusion is most likely correct, but there is no explanation given for why there would be more mucus in the cultured tracheas. Was the number of goblet cells increased during culture? Was the number of mucus-producing cells in the glands increased? Is the explanation instead that mucus is collecting on the cultured trachea that would be removed by mucociliary clearance in the live animal? This deserves some further discussion.

R2. These are excellent points, and we agree with your reasoning. We do not believe that the ex vivo tracheae have increased mucus producing cells or glands. We believe that the increase in mucus is solely due to its accumulation on the ex vivo trachea in the ex vivo condition. In an in vivo setting, mucus is swallowed or expectorated whereas the ex vivo mucus can pool at the end of the trachea (at least until it accumulates sufficient to be pushed off the end of the trachea, a phenomenon we have observed. We briefly discuss this in the beginning of this discussion paragraph (line 511), and we agree that an additional sentence after discussing the PAS results helps to clarify this point. To address this in text, we have added the following insert:

INSERT: “Although the mucus yield from a single ex vivo collection is already higher than a single collection from fresh tracheae, likely due to mucus accumulation that would normally be cleared in an in vivo setting, the ex vivo mucus yield may be substantially increased by repeated collections on the same trachea, a practice not possible on freshly excised tissue.”

Reviewer #2: Harris et al. have set out a rat trachea model to study the abnormalities of the CF respiratory tract ex vivo. The paper describes a convincing, very carefully detailed method to isolate and maintain tracheae for ex vivo experimentation and drug testing.

Main concerns:

1. Methods, Lines 180-185: This part can sound a bit confusing for someone who is not well acquainted with the topic. Have the whole tracheae undergone fixation and paraffinization for histopathology or just a part of each trachea? Also, step 4 says that µOCT was performed in freshly excised tracheae, which, I suppose, had not yet been processed for histological analysis. Kindly clarify the order in which these analyses – histopathology and imaging – were performed.

R1. This is a good point, and we agree that it needs clarification in text. We used the medial section of the trachea for histopathology. Furthermore, the tracheae used for histopathology were not used for any other analysis. The 21-day uOCT study from Figure 2 was longitudinal, so tracheae were used only for uOCT. We did not perform histopathology on trachea from figure 3 (WT v CF), to avoid any possible artifact by carbachol stimulation. To address this, we added the following inserts into the methods:

INSERT: “Cross sections of the medial trachea were immersion fixed in 10% neutral buffered formalin and submitted to histology laboratory for tissue processing, paraffin embedding and sectioning.”

INSERT: “Tracheae used for histopathology analysis were not used in any other analyses.”

2. Results: When downloaded, some figures have dark backgrounds, which make them difficult to see. Please make sure that all figures are visible in case the paper is sent for another review round.

R2. Thank you for bringing this to our attention. We will double check the figures in the submission portal and ensure their quality does not degrade upon submission.

3. Lines 293-295: The word “trend” does not sound as the most appropriate. In general, trend is used to describe a change over time or, if much, a pattern after repeated measurements with different sample. In the present study, only a few animals have been used, which is in accordance with ethical protocols, so the amount may not be representative enough to point out any trend.

R3. To more accurately present this data, we have changed the wording in text with the following insertion:

INSERT: “Histopathology of AB-PAS stained ex vivo trachea showed an increased number of goblet cells per length of epithelium in the CFTR−/− tracheae (0.015 ± 0.006 cells/mm) compared to WT (0.005 ± 0.002 cells/mm; P=0.19; Supplemental Figure 2), although these differences were not statistically signficant, likely due to high variance in CFTR−/− tracheae”

4. Figure 2: Were the experiments in Figure 2 conducted with CFTR-/- tissue or with WT tissue? If conducted with WT tissue, how long were the CFTR-/- tracheae viable?

R4. This is a good question. For the longitudinal studies in Figure 2, we used an equal number of WT and CF tracheae for the ex vivo and fresh tracheae groups. There were no differences in cilia coverage or CBF at any timepoint between the CF and WT trachea, so they were both included in the figure 2 comparisons. This is an important detail that should be included in the revised manuscript, and I appreciate you bringing it to our attention. To address this, we have added the following insert into the text:

INSERT: “An equal number of WT and CFTR−/ ex vivo tracheae were explanted, given 5 days to equilibrate, and imaged for cilia function analysis at day 5, 9, 14, and 21 to assess durability and stability. Fresh tracheae, with an equal number from WT and CFTR−/− rats, were harvested and imaged immediately upon explant for comparison, as previously described (16)”

5. Discussion: Please elaborate on the added value, difference and importance of this model compared to precision cut lung slices.

R5. This is an important and very comparable model to the one presented in this manuscript. To address this comparison, we added the following insert into a limitations paragraph in the discussion (see also R7, below).

INSERT: “A comparable ex vivo model, precision cut lung slices (PCLS), has emerged as a versatile tool to study lung tissue function and outcomes within the context of native cell populations and resident immune cells across multiple regions of the lung. Although a valuable tool, PCLS are limited in their utility to evaluate lung physiology at ALI, particularly mucociliary physiology in the context of a larger tissue, due to their fixation by agarose inflation and sampling of a small tissue area with inherent inconsistencies of native cell populations due to the extensive heterogeneity of the lung (45). The use of PCLS in conjunction with ex vivo tissue cultures may provide a powerful combination to comprehensively study respiratory responses in specific tissues as well as at the ALI.”

6. Please elaborate (discussion) on the contribution of trachea impairment in CF compared to the rest of the lung.

R6. We agree that this is an important thing to note when describing the limitations of the ex vivo trachea model. To address this in text, we added the following insert into the limitations paragraph of the discussion.

INSERT: “The ex vivo tracheae model is limited due to the sampling of only the tracheae, which excludes other regions of the lung that may be important for studying lung outcomes in CF or other diseases; however, the trachea has been widely utilized as an effective indicator of airway physiology outcomes in multiple disease states and animal models, due to its properties of fluid secretion and mucus production in glands and epithelia (12, 16, 44).”

Minor concerns:

1. This is merely technicality, but, overall, the authors mix the Results section with data interpretation, which, in turn, should be in the Discussion section. I would advise the authors to order the results in terms of what the authors consider most important, only with the description of the results. Likewise, the authors can interpret these findings in the Discussion section by following the same order in the result. This can make it easier to the reader to follow the authors’ train of thought.

R1: The revised manuscript allows the reader to understand the importance of results, using an occasion phrase demonstrating interpretation, when provided in context. Further delineation of complex interpretations is provided in the Discussion, as suggested.

2. Please avoid repetitive statements in the M&M (e.g., first sentence of 2.1 and 2.2).

R2. To avoid repetitive statements int the M&M, we removed the following two sentences from the text:

Deleted sentence: “Prepare culture cassettes (1 per trachea) at least 1 day prior to explanting.”

Deleted sentence: “Prepare and sterilize all components as necessary prior to harvest.”

3. The quality of the representative image of Figure S2 of the WT animal is not comparable to the CFTR-/- animal and should be replaced.

R3. We have replaced the WT representative with an epithelial section that better highlights goblet cells and is more comparable to the CFTR representative. Thank you for this suggestion.

4. Include full names of abbreviations at the end of every figure legend as figure should be able to stand alone without reading the main text.

R4. We have added abbreviations to the end of each figure legend. 

5. Please specify whether the ducts analyzed for Figure 4 were analyzed in the same trachea region (proximal vs. distal).

R5. The trachea glands and ducts were evaluated over the entire length of the trachea. We have added this in the methods with the following insert:

INSERT: “Glands and ducts were evaluated over the whole length of the trachea.”

6. Line 325: small typing error “previously” should read “previous”

R6. We have fixed this grammatical error in the text. 

7. For Figure 4: please provide a supplemental figure showing clear ducts in both WT and CFTR-/- tracheae after the washing protocol.

R7. We have not ascertained the ability for washing to clear adherent mucus within the gland ducts.

Reviewer #3: Introduction

Lines 60- 61, page 9 – the statement “and not easily amenable to genetic modification or longitudinal studies” should be softened. There are several genetically modified pigs and ferrets available. The demand for genetically modified models in these species may be less, but that doesn’t mean they are not amenable to genetic modification. Similarly, there several longitudinal studies in ferrets and pigs. It is recommended that this statement be softened or re-worded.

R1. We agree that this statement needs to be reworded, especially given its context in the introduction. We have changed the text to the following:

INSERT: “Although these species have proven useful for studying mucus pathophysiology and to advance therapeutic strategies (9, 12, 14), they can be costly and difficult to maintain.”

Methods

2.4 – is the washing procedure adequate to remove mucus from the CFTR-/- rat airway? For example, mucus is likely more adherent in CFTR-/- rat airways.

R2. We have found the wash protocol to be adequate for removing mucus from the CF rat airway. We were able to check this via uOCT after washing, although we did not collect data for this. To better highlight this in text, we added the following insert into the methods.

INSERT: “When performing studies on mucus, ensure that mucus is adequately removed from the tracheae prior to experimentation. Although DPBS washes were sufficient to remove accumulated mucus in the following studies, washes with reducing agents such as 3mM dithiothreitol, as described on HBEs, may need to be implemented if mucus remains after multiple DBPS washes (21).”

2.5 – has the equation listed been validated? Meaning, have the expected concentrations of drugs using that equation been measured in the media and/or in the tissues?

R3. The equation has not been experimentally validated, other than we have confirmed that the surgifoam holds the volume of liquid that we calculated by weighing while saturated and after desiccation. The equation is based off of the assumption that any drug concentrations added to a cassette will be diluted by the volume of media already contained in the pre-saturated sponge. We have provided guidance in the methods with the following insert:

INSERT: “Although not performed in the following studies, further validation of actual drug concentrations in the media following this step may be necessary depending on the study being performed.”

6.0 PAS Slot Blot – Is it assumed that PAS is only detecting secreted mucins and not other glycoproteins or glycolipids? Additionally, PAS since only detects neutral mucins, are conclusions draw about secreted mucins accurate (since some literature suggests that mucins in CF may be more acidic in form, and would be detected by Alcian blue preferentially.

R4. This is a good point to address. We are assuming that the PAS staining is stemming from mucin, as we evaluated secreted mucus collections from the lumen. Although it is possible that other proteins containing glycols were stained, this would still indicate increased presence of mucus secretions. We chose PAS as the mucin detection method based on its ability to stain neutral and acidic mucins, as we wanted to evaluate total mucin content, and mucin acidity may be altered in different diseased states. The PAS method ensured that all mucin, neutral and acidic, were stained in a mechanism that is not dependent on acidic moieties. 

Results

Were sex differences noted?

R5. Tracheae were explanted from male and female rats (similar numbers of each) in these studies and matched accordingly. There were no noted sex differences in any outcomes. To clarify this in text, the following insert was added to the methods:

INSERT: “All experimental groups were matched by age and sex, with no noted sex differences in any outcomes.

The images in Figure 2 for ex vivo trachea give impression there is more vacuoles in the trachea – is that a true observation or just due to the image selected?

R6. There were no noted differences in vacuole numbers. Any appearance of this is likely due to processing artifacts and/or the image selected for representation. We chose images that we believed best represented the epithelium and submucosa. 

Discussion

Nicely written. A recommendation is to include a limitations paragraph. For example, one limitation is that ex vivo preps do not allow for studying the interaction of the epithelia with the immune cells, unless immune cells are added. Additionally, though cholinergic responses remained intact, there will be a loss of extrinsic neural innervation, which is important for surface epithelia and submucosal gland function.

R7. We agree that this is an important section to be included in the manuscript. We added a limitations paragraph, and aimed to address these points as well as others made by reviewer two in the context of other comparable ex vivo models. 

INSERT: “The ex vivo tracheae model is limited due to the sampling of only the tracheae, which excludes other regions of the lung that may be important for studying lung outcomes in CF or other diseases; however, the trachea has been widely utilized as an effective indicator of airway physiology outcomes in multiple disease states and animal models, due to its properties of fluid secretion and mucus production in glands and epithelia (12, 16, 44). A comparable ex vivo model, Precision cut lung slices (PCLS), has emerged as a versatile tool to study lung tissue function and outcomes within the context of native cell populations and resident immune cells across multiple regions of the lung. Although a valuable tool, PCLS are limited in their utility to evaluate lung physiology at ALI, particularly mucociliary physiology in the context of a larger tissue, due to their fixation by agarose inflation and sampling of a small tissue area with inherent inconsistencies of native cell populations due to the extensive heterogeneity of the lung (45). The use of PCLS in conjunction with ex vivo tissue cultures may provide a powerful combination to comprehensively study respiratory responses in specific tissues as well as at the ALI. Another limitation to consider when using the ex vivo trachea model is the absence of non-resident immune cells, unless added extrinsically to the cultures. This may be an important consideration when studying airway physiology outcomes that may rely on the presence of immune cells. Additionally, although we show that cholinergic stimulation remains intact after 7 days post explant (Supplemental Figure 3), there will likely be a loss of extrinsic neural innervation, which has been shown to be important for epithelial and submucosal gland function (46).”

---

## [Editor Report · Decision Letter 1]

11 Oct 2023

An ex vivo rat trachea model reveals abnormal airway physiology and a gland secretion defect in cystic fibrosis

PONE-D-23-16864R1

Dear Dr. Steven M. Rowe,

We’re pleased to inform you that your manuscript has been judged scientifically suitable for publication and will be formally accepted for publication once it meets all outstanding technical requirements.

Kind regards,

Abdelwahab Omri, Pharm B, Ph.D, Laurentian University, Canada

Academic Editor

PLOS ONE

---

## [Editor Report · Acceptance letter]

16 Oct 2023

PONE-D-23-16864R1 

An *ex vivo* rat trachea model reveals abnormal airway physiology and a gland secretion defect in cystic fibrosis 

Dear Dr. Rowe:

I'm pleased to inform you that your manuscript has been deemed suitable for publication in PLOS ONE. Congratulations! Your manuscript is now with our production department. 

Kind regards, 

on behalf of

Dr. Abdelwahab Omri 

Academic Editor

PLOS ONE